# Enhanced Meta Reinforcement Learning using Demonstrations in Sparse Reward Environments

**Desik Rengarajan**[*]  **Sapana Chaudhary**[*]
**Jaewon Kim**   **Dileep Kalathil**   **Srinivas Shakkottai**
Department of Electrical and Computer Engineering, Texas A&M University
{desik,sapanac,jwkim8804,dileep.kalathil,sshakkot}@tamu.edu

## Abstract

Meta reinforcement learning (Meta-RL) is an approach wherein the experience gained from solving a variety of tasks is distilled into a meta-policy. The meta-policy, when adapted over only a small (or just a single) number of steps, is able to perform near-optimally on a new, related task. However, a major challenge to adopting this approach to solve real-world problems is that they are often associated with sparse reward functions that only indicate whether a task is completed partially or fully. We consider the situation where some data, possibly generated by a sub-optimal agent, is available for each task. We then develop a class of algorithms entitled Enhanced Meta-RL using Demonstrations (EMRLD) that exploit this information—even if sub-optimal—to obtain guidance during training. We show how EMRLD jointly utilizes RL and supervised learning over the offline data to generate a meta-policy that demonstrates monotone performance improvements. We also develop a warm started variant called EMRLD-WS that is particularly efficient for sub-optimal demonstration data. Finally, we show that our EMRLD algorithms significantly outperform existing approaches in a variety of sparse reward environments, including that of a mobile robot.

## 1   Introduction

Meta-Reinforcement Learning (meta-RL) is an approach towards quickly solving similar tasks while only gathering a few samples for each task. The fundamental idea behind approaches such as the popular Model Agnostic Meta-Learning (MAML) [9] is to determine a universal meta-policy by combining the information gained over working on several tasks. This meta-policy is optimized in a manner such that, when adapted using a small number of samples gathered for a given task, it is able to perform near optimally on that specific task. While the dependence of MAML on only a small number of samples for task-specific adaptation is very attractive, its also means that these samples must be highly representative of that task for meaningful adaptation and meta-learning. However, in many real-world problems, rewards provided are sparse in that they might only provide limited feedback. For instance, there might be a reward only if a robot gets close to designated way point, with no reward otherwise. Hence, MAML is likely to fail on both counts of task-specific adaptation and optimization of the meta-policy over tasks in sparse reward environments. Making progress towards learning a viable meta-policy in such settings is challenging without additional information.

Many real-world tasks are associated with empirically determined policies used in practice. Such policies could be inexpert, but even limited demonstration data gathered from applying these policies could contain valuable information in the sparse reward context. While the fact that the policy generating the data could be inexpert suggests that direct imitation might not be optimal, supervised

---

[*]Equal contribution.

learning over demonstration data could be used for *enhancing* adaptation and learning. How best should we use demonstration data to enhance the process of meta-RL in the sparse reward setting?

Our goal is a principled design of a class of meta-RL algorithms that can exploit demonstrations from inexpert policies in the sparse reward setting. Our general approach follows two-step algorithms like MAML that employ: (i) Task-specific Adaptation: execute the current meta-policy on a task and adapt it based on the samples gathered to obtain a task-specific policy, and (ii) Meta-policy Optimization: execute task-specific policies on an ensemble of tasks to which they are adapted, and use the samples gathered to optimize the meta-policy from whence they were adapted. Our key insight is that we can enhance RL-based policy adaptation with behavior cloning of the inexpert policy to guide task-specific adaptation in the right direction when rewards are sparse. Furthermore, execution of such an enhanced adapted policy should yield an informative sample set that indicates how best to obtain the sparse rewards for meta-policy optimization. We aim at analytically and empirically capturing this progression of policy improvement starting from task-specific policy adaption to the ultimate meta-policy optimization. Thus, as long the inexpert policy has an advantage, we must be able to exploit it for meta-policy optimization.

Our main contributions are as follows. (i) We derive a policy improvement guarantee result for MAML-like two-step meta-RL algorithms. We show that the inclusion of demonstration data can further increase the policy improvement bound as long as the inexpert policy that generated the data has an advantage. (ii) We propose an approach entitled Enhanced Meta-RL using Demonstrations (EMRLD) that combines RL-based policy improvement and behavior cloning from demonstrations for task-specific adaptation. We further observe that directly applying the meta-policy to a new sparse-reward task sometimes does not yield informative samples, and a warm-start to the meta-policy using the demonstrations significantly improves the quality of the samples, resulting in a variant that we call EMRLD-WS. (iii) We show on standard MuJoCo and two-wheeled robot environments that our algorithms work exceptionally well, even when only provided with just one trajectory of sub-optimal demonstration data per task. Additionally, the algorithms work well even when exposed only to a small number of tasks for meta-policy optimization. (iv) Our approach is amenable to a variety of meta-RL problems wherein tasks can be distinguished across rewards (e.g., whether forward or backward motion yields a reward for a task) or across environment dynamics (e.g., the amount of environmental drift that a wheeled robot experiences changes across tasks). To illustrate the versatility of EMRLD, we not only show simulations on different continuous control multi-task environments, but also demonstrate its excellent performance via real-world experiments on a TurtleBot robot [2]. We provide videos of the robot experiments and code at `https://github.com/DesikRengarajan/EMRLD`.

**Related Work:** Here, we provide a brief overview of the related works. We leave a more thorough discussion on related works to the Appendix.

**Meta learning:** Basic ideas on the meta-learning framework are discussed in [15, 34, 8]. Model-agnostic meta-learning (MAML) [9] introduced the two-step approach described above, and can be used in the supervised learning and RL contexts. However, in its native form, the RL variant of MAML can suffer from issues of inefficient gradient estimation, exploration, and dependence on a rich reward function. Among others, algorithms like ProMP [28] and DiCE [11] address the issue of inefficient gradient estimation. Similarly, E-MAML [1, 33] and MAESN [12] deal with the issue of exploration in meta-RL. PEARL [26] takes a different approach to meta-RL, wherein task specific contexts are learnt during training, and interpreted from trajectories during testing to solve the task. HTR [23] relabels the experience replay data of any off-policy algorithm such as PEARL [26] to overcome exploration difficulties in sparse reward goal reaching environments. Different from this approach, we use demonstration data to aid learning and are not restricted to goal reaching tasks.

**RL with demonstration:** Leveraging demonstrations is an attractive approach to aid learning [13, 36, 21]. Earlier work has incorporated data from both expert and inexpert policies to assist with policy learning in sparse reward environments [21, 14, 36, 17, 27]. In particular, [14] utilizes demonstration data by adding it to the replay buffer for Q-learning, [25] proposes an online fine-tuning algorithm by combining policy gradient and behavior cloning, while [27] proposes a two-step guidance approach where demonstration data is used to guide the policy in the initial phase of learning.

**Meta-RL with demonstration:** Meta Imitation Learning [10] extends MAML [9] to imitation learning from expert video demonstrations. WTL [40] uses demonstrations to generate an exploration algorithm, and uses the exploration data along with demonstration data to solve the task. GMPS [20]

extends MAML [9] to leverage expert demonstration data by performing meta-policy optimization via supervised learning. Closest to our approach are GMPS [20] and Meta Imitation Learning [10], and we will focus on comparisons with versions of these algorithms, along with the original MAML [9].

Our work differs from prior work on meta-RL with demonstration in several ways. First, existing works assume the availability of data generated by an expert policy, which severely limits their ability to improve beyond the quality of the policy that generated the data. This degrades their performance significantly when they are presented with sub-optimal data generated by an inexpert policy that might be used in practice. Second, these works use demonstration data in a purely supervised learning manner, without exploiting the RL structure. We use a combination of loss functions associated with RL and supervised learning to aid the RL policy gradient, which enables our approach to utilize any reward information available. This makes it superior to existing work in the sparse reward environment, which we illustrate in several simulation settings and real-world robot experiments.

## 2   Preliminaries

A Markov Decision Processes (MDP) is typically represented as a tuple $< \mathcal{S}, \mathcal{A}, R, P, \gamma, \rho >$, where $\mathcal{S}$ is the state space, $\mathcal{A}$ is the action space, $R : \mathcal{S} \times \mathcal{A} \to \mathbb{R}$ is the reward function, $P : \mathcal{S} \times \mathcal{A} \to \Delta(\mathcal{S})$ is the transition probability function, $\gamma$ is the discount factor, and $\rho \in \Delta(\mathcal{S})$ is the initial state distribution. The value function $V^\pi$ and the state-action value function $Q^\pi$ of a policy $\pi$ are defined as $V^\pi(s) = \mathbb{E}\left[\sum_{t=0}^\infty \gamma^t R(s_t, a_t) | s_0 = s\right]$ and $Q^\pi(s, a) = \mathbb{E}\left[\sum_{t=0}^\infty \gamma^t R(s_t, a_t) | s_0 = s, a_0 = a\right]$. The advantage function $A^\pi$ is defined as $A^\pi(s, a) = Q^\pi(s, a) - V^\pi(s)$. A policy $\pi$ generates a trajectory $\tau$ where $\tau = (s_0, a_0, s_1, a_1, \dots), s_0 \sim \rho, a_t \sim \pi(s_t, \cdot), s_{t+1} \sim P(\cdot|s_t, a_t)$. Since the randomness of $\tau$ is specified by $P$ and $\pi$, we denote it as $\tau \sim (P, \pi)$. The goal of a reinforcement learning algorithm is to learn a policy that maximizes the expected infinite horizon discounted reward defined as $J(\pi) = \mathbb{E}_{\tau \sim (\pi, P)}\left[\sum_{t=0}^\infty \gamma^t R(s_t, a_t)\right]$. It is easy to see that $J(\pi) = \mathbb{E}_{s_0 \sim \rho}[V^\pi(s_0)]$.

The discounted state-action visitation frequency of policy $\pi$ is defined as $d^\pi(s, a) = (1 - \gamma) \sum_{t=0}^\infty \gamma^t \mathbb{P}(s_t = s, a_t = a), s_0 \sim \rho$. The discounted state visitation frequency is defined as the marginal $d^\pi(s) = \sum_{a \in \mathcal{A}} d^\pi(s, a)$. It is straightforward to see that $d^\pi(s, a) = \pi(s, a) d^\pi(s)$.

The total variation (TV) distance between two distributions $p$ and $q$ is defined as $D_{\text{TV}}(p, q) = (1/2) \sum_x |p(x) - q(x)|$, the average TV distance between two policies $\pi_1$ and $\pi_2$, averaged w.r.t. the $d^{\pi_\pi}$ is defined as $D_{\text{TV}}^\pi(\pi_1, \pi_2) = \mathbb{E}_{s \sim d^\pi}[D_{\text{TV}}(\pi_1, \pi_2)]$.

**Gradient-based meta-learning:** The goal of a meta-learning algorithm is to learn to perform optimally in a new (testing) task using only limited data, by leveraging the experience (data) from similar (training) tasks seen during training. Gradient-based meta-learning algorithms achieve this goal by learning a meta-parameter which will yield a good task specific parameter after performing only a few gradient steps w.r.t. the task specific loss function using the limited task specific data.

Meta-learning algorithms consider a set of tasks $\mathcal{T}$ with a distribution $p$ over $\mathcal{T}$. Each task $i \sim p(\mathcal{T})$ is also associated with a data set $\mathcal{D}_i$, which is typically divided into training data $\mathcal{D}_i^{\text{tr}}$ used for task specific adaptation and validation data $\mathcal{D}_i^{\text{val}}$ used for meta-parameter update. The objective of the gradient-based meta-learning is typically formulated as

$$\min_\theta \ \mathbb{E}_{i \sim p(\mathcal{T})} \left[ \mathcal{L}_i \left( \theta - \alpha \nabla_\theta \mathcal{L}_i(\theta, \mathcal{D}_i^{\text{tr}}), \mathcal{D}_i^{\text{val}} \right) \right], \tag{1}$$

where $\mathcal{L}_i$ is the loss function corresponding to task $i$ and $\alpha$ is the learning rate. Here, $\theta_i(\theta) = \theta - \alpha \nabla_\theta \mathcal{L}_i(\theta, \mathcal{D}_i^{\text{tr}})$ is the task specific parameter obtained by one step gradient update starting from the meta-parameter $\theta$, and the goal is to find the best meta-parameter which will minimize the meta loss function $\mathcal{L}(\theta) = \mathbb{E}_{i \sim p(\mathcal{T})}[\mathcal{L}_i(\theta_i(\theta))]$

**Gradient-based meta-reinforcement learning:** The gradient-based meta-learning framework is applicable both in supervised learning and reinforcement learning. In RL, each task $i$ corresponds to an MDP with task specific model $P_i$ and reward function $R_i$. We assume that the state-action spaces are uniform across the tasks, thus ensuring the first level of task similarity. Task specific data $\mathcal{D}_i$ is the trajectories $\tau_{i,m} = (s_0^m, a_0^m, s_1^m, a_1^m, \dots), s_0 \sim \rho, a_t \sim \pi(s_t, \cdot), s_{t+1} \sim P_i(\cdot|s_t, a_t)$ for $1 \le m \le M$ generated according to some policy $\pi$. Since the randomness of the trajectory $\tau_{i,m}$ is specified by $P_i$ and $\pi$, we denote it as $\tau_{i,m} \sim (P_i, \pi)$. We consider the function approximation setting where each policy $\pi$ is represented by a function parameterized by $\theta \subset \Theta$ and is denoted as

$\pi_\theta$. The task specific loss in meta-RL is defined as $\mathcal{L}_{\mathrm{RL}}^i(\theta) = -J_i(\pi_\theta)$. The gradient $\nabla_\theta \mathcal{L}_{\mathrm{RL}}^i(\theta)$ can then be computed using policy gradient theorem.

The standard meta-RL training is done follows. A task $i \sim p(\mathcal{T})$ (usually a batch of tasks) is sampled at each iterate $k$ of the algorithm. Now, starting with meta-parameter $\theta_k$, the training data $\mathcal{D}_i^{\mathrm{tr}}$ for task adaptation is generated as the trajectories $(\tau_{i,m})_{m=1}^M$, where $\tau_{i,m} \sim (P_i, \pi_{\theta_k})$, and the updated parameter $\theta_{i,k}$ for task $i$ is computed by policy gradient evaluated on $\mathcal{D}_i^{\mathrm{tr}}$. The validation data $\mathcal{D}_i^{\mathrm{val}}$ is then collected as trajectories $(\tau_{i,m})_{m=1}^M$, where $\tau_{i,m} \sim (P_i, \pi_{\theta_{i,k}})$, and the meta-parameter $\theta_k$ is updated by policy gradient evaluated on $\mathcal{D}_i^{\mathrm{val}}$. In the next section, we will introduce a modified approach which will leverage the demonstration data for task adaptation and meta-parameter update.

## 3 Meta-RL using Demonstration Data

Most gradient-based meta-RL algorithms learn the optimal meta-parameter and the task-specific parameter from scratch using on-policy approaches. These algorithms exclusively rely on the reward feedback obtained from the training and validation data trajectories collected through the on-policy roll-outs of the meta-policy and task-specific policy. However, in RL problems with sparse rewards, a non-zero reward is typically achieved only when the task is completed or near-completed. In such sparse rewards settings, trajectories generated according to a policy that is still learning may not achieve any useful reward feedback, especially in the early phase of learning. In other words, since the reward feedback is zero or near-zero, the policy gradient will also be similar, resulting in non-meaningful improvement in the policy. Hence, standard meta-RL algorithms such as MAML, which rely crucially on reward feedback, will not be able to make much progress towards learning a valuable task-specific or meta-policy in sparse reward settings.

Learning the optimal control policy in sparse reward environments has been recognized as a challenging problem even in the standard RL setting, since most state-of-the-art RL algorithms fail to learn any meaningful policies even after a large number of training episodes [25, 17, 27]. One widely accepted approach to overcome this challenge is known as *learning from demonstration*, wherein demonstration data obtained from an expert [25] or inexpert policy [17, 27] is used to aid online learning. The intuitive idea is that, even though the demonstration data does not contain any reward feedback, it can be used to guide the learning agent to reach non-zero reward regions of state-action spaces. This guidance, usually in the direction of the goal/target, is achieved by inferring some pseudo reward signal through supervised learning approaches using demonstration data.

*Can we enhance the performance of meta-RL algorithms in sparse reward environments by using demonstration data from sub-optimal experts?* Meta-RL in sparse reward environments is significantly more challenging than that of the standard RL setting. This is because the reward feedback serves the dual objectives of adapting the meta-parameter to specific tasks and for updating the meta-parameter itself. We note that demonstration data helps with both of these objectives. Firstly, use of demonstration data to guide task-specific adaptation becomes important because adaptation is achieved in one or a few gradient steps, and policy resulting from each adaptation step might not achieve meaningful reward in a sparse reward setting. Secondly, making use of demonstration data for meta-parameter update is equally important because of the role of meta-policy as a reward-yielding exploratory policy. Intuitively, the meta-policy should yield trajectories that reach in the vicinity of the reward-achieving region of the state-action spaces. This does not happen in sparse reward environments. However, using the guidance from demonstration data, the task-specific policy obtained after the task adaptation may be able to generate trajectories that will reach within the reward-achieving region resulting in performance acceleration of the meta-policy.

For meta-learning with demonstration, we assume that each task $i$ is associated with demonstration data $\mathcal{D}_i^{\mathrm{dem}}$, which contains a trajectory generated according to a demonstration policy $\pi^{\mathrm{dem}}$ in an environment with model $P_i$. We *do not assume* that $\pi^{\mathrm{dem}}$ is the optimal policy for task $i$ because in many real-world applications $\mathcal{D}_i^{\mathrm{dem}}$ could be generated using an inexpert policy. Our key idea is to enhance task adaptation using the demonstration data by introducing an additional gradient term corresponding to the supervised learning guidance loss. We define the supervised learning loss function for task $i$ as $\mathcal{L}_i^{\mathrm{BC}}(\theta, \mathcal{D}_i^{\mathrm{dem}}) = -\sum_{(s,a) \in \mathcal{D}_i^{\mathrm{dem}}} \log \pi_\theta(a|s)$. We note that, though this loss function is the same as in behavior cloning (BC), we use it directly in the gradient update instead of performing a simple warm start. This approach is known to achieve superior performance than naive BC warm starting in standard RL problem under the sparse reward setting [25, 17, 27]. The

task adaptation step at iteration $k$, starting with meta-parameter $\theta_k$ is now obtained as

$$\theta_{k,i} = \theta_k - \alpha \nabla_\theta \left( w_{rl} \mathcal{L}_i^{RL}(\theta; \mathcal{D}_i^{tr}) + w_{bc} \mathcal{L}_i^{BC}(\theta, \mathcal{D}_i^{dem}) \right) |_{\theta=\theta_k}, \tag{2}$$

where $w_{rl}$ and $w_{bc}$ are hyperparameters that control the extant to which RL and demonstration data influence the gradient.

The next question is: how do we use demonstration data in the meta-parameter update? One approach is to use only the demonstration data with a supervised learning loss function for updating the meta-parameter as done in [20]. We conjecture that such a reduction to supervised learning will severely limit the learning capability of the algorithm. Firstly, if demonstration data is obtained from an inexpert policy, this approach will never be able to achieve the optimal performance. This is because the role of the meta-policy as a reward-yielding exploratory policy will be limited by true performance of the inexpert policy. Secondly, the task-specific policies obtained according to (2) may be able to reach within the reward-yielding region of state-action space as we mentioned before. Hence, the validation data $\mathcal{D}_i^{val}$ collected through the roll-out of the policies obtained after task adaptation might contain extremely valuable reward feedback. Utilizing this data could potentially have a significant impact on improving the learning of the meta-parameter. Thus, in our approach, we update the meta-parameter using the RL loss with policy gradient as follows.

$$\theta_{k+1} = \theta_k - \beta \nabla_\theta \sum_i \mathcal{L}_i^{RL}(\theta - \alpha \nabla_\theta \left( w_{rl} \mathcal{L}_i^{RL}(\theta; \mathcal{D}_i^{tr}) + w_{bc} \mathcal{L}_i^{BC}(\theta, \mathcal{D}_i^{dem}) \right); \mathcal{D}_i^{val}) \tag{3}$$

We note that the demonstration data is indeed used in the meta-parameter update implicitly, as its impact can be observed in (3). We found empirically that the double use of the demonstration data, either by adding an additional gradient through a BC loss function, or by replacing $\mathcal{D}_i^{val}$ with $\mathcal{D}_i^{dem}$ results in similar or worse performance than the approach described above.

We now formally present our algorithm called Enhanced Meta-RL using Demonstrations (EMRLD).

---

**Algorithm 1** Enhanced Meta-RL using Demonstrations (EMRLD)

---

1: **Input:** Set $\mathcal{T}$ of $N$ tasks, demonstration data $\mathcal{D}_i^{dem}$ for task $i = 1, \ldots, N$
2: **Input:** Adaptation learning rate $\alpha$, meta-learning rate $\beta$
3: Initialize meta parameter $\theta_0$
4: **for** $k = 0, 1, \ldots$ **do**
5:     **for** $i \in \mathcal{T}$ **do**
6:         Execute the meta policy $\pi_{\theta_k}$ for task $i$ to collect $\mathcal{D}_i^{tr}$
7:         **Task adaptation:** Compute task adapted parameter $\theta_{k,i}$ using $\mathcal{D}_i^{tr}$ and $\mathcal{D}_i^{dem}$ according to 2
8:         Collect data $\mathcal{D}_i^{val}$ by executing adapted policy $\pi_{\theta_{k,i}}$ for meta-policy update
9:     **end for**
10:     **Meta-parameter update:** Update meta-policy according to 3 using $\mathcal{D}_i^{val}$
11: **end for**

---

We now present a theoretical justification of why EMRLD should have a superior performance in the sparse reward setting as compared to other gradient based algorithms that do not use demonstration data. First, we introduce some notation. Let $\pi_k = \pi_{\theta_k}$ be the meta-policy used at iteration $k$ of our algorithm. Also, let $\pi_{k,i}$ be the policy obtained after task-specific adaptation for task $i$. Recall that $J_i(\pi_{k,i})$ is the value of the policy for the MDP corresponding to task $i$. Similarly, we can define the state-action value function and advantage function of policy $\pi_{k,i}$ for task $i$ as $Q_i^{\pi_{k,i}}$ and $A_i^{\pi_{k,i}}$, respectively. Also, let $d_i^{\pi_{k,i}}$ be the visitation frequency of policy $\pi_{k,i}$ for task $i$. Now, we can define the value of the meta-policy $\pi_k$ over the ensemble of all tasks as $J_{meta}(\pi_k) = \mathbb{E}_{i \sim p(\mathcal{T})}[J_i(\pi_{k,i})]$.

If the demonstration data has to be useful, it should provide a reasonable amount of guidance. In particular, we would like the task-specific policy adapted using this data to collect feedback that would ensure good meta-policy updates, particularly in the initial stages of meta-training. Since the capability of the demonstration data to guide adaptation will depend on the demonstration policy $\pi_i^{dem}$ according to which it is generated, we make the following assumption about $\pi_i^{dem}$.

**Assumption 3.1.** During the initial stages of meta-training, $\mathbb{E}_{a \sim \pi_i^{dem}(s, \cdot)}[A_i^{\pi_{k,i}}(s, a)] \geq \Delta$, for all $s \in \mathcal{S}$ and $i \in \mathcal{T}$.

Assumption 3.1 implies that during the initial stages of meta-training, the demonstration policy can provide a higher advantage on average than the current policy adapted to that task. This is a reasonable assumption, since any reasonable demonstration policy is likely to perform much better than an untrained policy in the initial phase of learning. We also note that a similar assumption was used in learning from demonstration literature [17, 27]

We now present the performance improvement result for EMRLD.

**Theorem 3.2.** *Let $\pi_k = \pi_{\theta_k}$ be the meta-policy used at iteration $k$ of our algorithm and let $\pi_{k,i}$ be the policy obtained after task adaptation in task $i$. Let Assumption 3.1 holds for $\pi_k$. Then,*

$$
J_{meta}(\pi_{k+1}) - J_{meta}(\pi_k) \geq \left( \frac{1}{1-\gamma} \mathbb{E}_{i \sim p(\mathcal{T}),(s,a) \sim d_i^{\pi_{k,i}}} \left[ \frac{\pi_{k+1,i}(s,a)}{\pi_{k,i}(s,a)} A_i^{\pi_{k,i}}(s,a) \right] \right.
$$

$$
\left. - \frac{2C_1}{1-\gamma} \mathbb{E}_{i \sim p(\mathcal{T})} \left[ D_{TV}^{\pi_{k,i}} (\pi_{k+1,i}, \pi_{k,i}) \right] \right) + \left( \frac{\Delta}{1-\gamma} - \frac{2C_1}{1-\gamma} \mathbb{E}_{i \sim p(\mathcal{T})} \left[ D_{TV}^{\pi_{k+1,i}} \left( \pi_{k+1,i}, \pi_i^{dem} \right) \right] \right)
$$

Theorem 3.2 presents a lower bound for the meta policy improvement as a sum of two groups of terms. Maximizing the first term in group one with a constraint on its second term will ensure a higher lower bound and hence an improvement in the meta-parameter training. We notice that this is indeed achieved by the TRPO step used in the meta-parameter update. Hence, this first group is the same for any MAML-type of algorithm. The advantage of the demonstration data is revealed in the second group of terms. The term $\Delta/(1-\gamma)$ adds a positive quantity to the lower bound, and this contribution from this second group of term can be maximized by minimizing $\mathbb{E}_{i \sim p(\mathcal{T})}[D_{TV}^{\pi_{k+1,i}} \left( \pi_{k+1,i}, \pi_i^{dem} \right)]$. However, this minimization is hard to perform in practice because estimating $D_{TV}^{\pi_{k+1,i}}$ requires sampling the data according to $\pi_{k+1,i}$, and this is not feasible at iteration $k$. Hence, in practice, we replace that term by $\mathbb{E}_{i \sim \mathcal{T}}[D_{TV}^{\pi_i^{dem}} \left( \pi_{k+1,i}, \pi_i^{dem} \right)]$. This can be easily achieved by including the standard maximum likelihood objective in the adaptation step. Thus, EMRLD both exploits the advantage offered by an RL step, as well as that of behavior cloning for meta-policy optimization.

We can further improve the performance of EMRLD by including a behavior cloning warm starting step before performing the update (2). We simplify this warm start to a one step gradient as, $\theta_{\tilde{k},i} = \theta_k - \alpha \nabla_\theta \mathcal{L}_i^{BC}(\theta, \mathcal{D}_i^{dem})|_{\theta=\theta_k}$, and then do the task adaptation as in (2) starting with $\theta_{\tilde{k},i}$. We call this version of our algorithm as EMRLD-WS. Such a warm start is likely to provide more meaningful samples than directly rolling out the meta-policy to obtain samples for task-specific adaptation. In the next section, we will see empirically how our design choices for EMRLD and EMRLD-WS enable them to learn policies that provide higher rewards using only a small amount of (even sub-optimal) demonstration data.

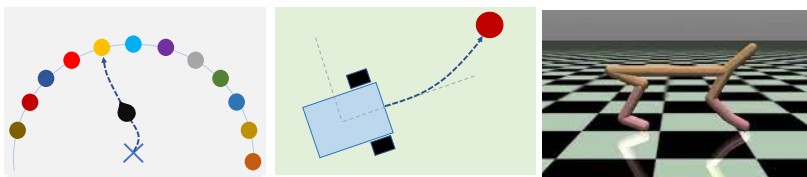

Figure 1: Point2D Navigation (first), Two Wheeled Locomotion (second) and HalfCheetah (last).

## 4 Experimental Evaluation

We evaluate the performance of EMRLD based on whether the meta-policy it generates is a good initial condition for task-specific adaptation in sparse-reward environments over (i) Tasks already seen in training and (ii) New unseen tasks. We seek to validate the conjecture that in the sparse-reward setting, EMRLD should be able to leverage even demonstrations of inexpert policies to attain high test performance over previously unseen tasks. We do so with regard to two classes of tasks, namely,

• Tasks that differ in their reward functions: Simulation experiments on Point2D Navigation [9], TwoWheeled Locomotion [12], and HalfCheetah [38, 35]

• Tasks that differ in the environment dynamics: Real-world experiments using a TurtleBot, which is a two-wheeled differential drive robot [2].

## 4.1 Experiments on simulated environments

**Sparse multi-task environments** We present simulation results for three standard environments shown in Figure 1 and described below. We train over a small number of tasks that differ in their reward functions. We generate unknown tasks for test by randomly modifying the reward function.

*Point2D Navigation* is a 2 dimensional goal-reaching environment. The states are the $(x, y)$ location of the agent on a 2D plane. The actions are appropriate 2D displacements $(dx, dy)$. Training tasks are defined by a fixed set of 12 goal locations on a semi-circle of radius 2. The agent is given a zero reward everywhere except when it is a certain distance near the goal location, making the reward function highly *sparse*. Within a single task, the objective of the agent is to reach the goal location in the least number of time steps starting from origin. Test tasks are generated by sampling any point on the semicircle as the goal.

*TwoWheeled Locomotion* environment is a goal-reaching with sparse rewards, similar to Point2D Navigation. However, the robot is constrained by the permissible actions (limits on angular and linear velocity) and trajectories feasible based on the turning radius of the robot. Here, our training tasks are a fixed set of 24 goal locations on a semi-circle of radius 2, while test goals are sampled randomly. Further details on state-space and dynamics are provided in the Appendix.

*HalfCheetah Forward-Backward* consists of two tasks in which the HalfCheetah agent learns to either move in the forward (task 1) or backward (task 2) directions with as high velocity as possible. The agent gets a reward only after it has moved a certain number of units along the x-axis in the correct direction, making the rewards sparse. Training and test are under the same two tasks.

**Optimal data and sub-optimal data** We provide a limited amount of demonstration data in the form of *just one trajectory per task for guidance*. Optimal data consists of $(s, a)$ transitions generated by an expert policy trained using TRPO. Sub-optimal data is generated by an inexpert, partially trained TRPO policy with induced action noise and truncated trajectories as shown in Figure 2.

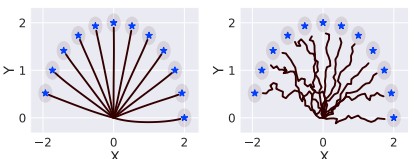

Figure 2: Optimal and Sub-optimal demonstrations for Point2D Navigation.

**Baselines** We compare the performance of our algorithm against the following gradient based meta-reinforcement learning algorithms: (i) **MAML:** [9] The standard MAML algorithm for meta-RL (ii) **Meta-BC:** A variant of [10]; this is a supervised learning/behavior cloning version of MAML, where the maximum likelihood loss is used in the adaptation as well as the meta-optimization steps. (iii) **GMPS:** Guided meta policy search [20], which uses RL for gradient based adaptation, and uses demonstration data for supervised meta-parameter optimization. The implemention of our algorithms and baselines is based on a publicly available meta-learning code base [3] licensed under the MIT License.

**Performance with optimal demonstration data:** We illustrate the training and testing performance of the different algorithms trained and tested with optimal data in Figure 3. The top row of Figure 3 shows the average adapted return across training tasks of the meta-policy during training iterations. The bottom row of Figure 3 shows the average return of the trained meta-policy adapted across testing tasks over adaptation steps. We see that our algorithms out perform the others by obtaining the highest average return, and are able to quickly adapt to testing tasks with just one adaptation step and one trajectory of demonstration data. Additionally, our algorithms demonstrate a nearly-monotone improvement in average return demonstrating stable learning. Meta-BC fails and has unstable training performance as the amount of demonstration data available per task is very small. Training over only a small number of tasks further hampers the performance of Meta-BC. MAML and GMPS fail to learn due to sparsity of the environment as the purely RL adaptation step incurs almost zero reward, and hence, negligible learning signal. Furthermore, GMPS is hampered in the meta update step due to availability of only a small amount of demonstration data per-task.

**Performance with sub-optimal demonstration data:** EMRLD uses a combination of RL and imitation, which is valuable when presented with sub-optimal demonstrations. For the Point2D Navigation environment, we collect sub-optimal data for each task using a partially trained agent with induced action noise, and truncate the trajectories short of the reward region. Hence, pure imitation cannot reach the goal. For the TwoWheeled Locomotion environment, we collect data in a

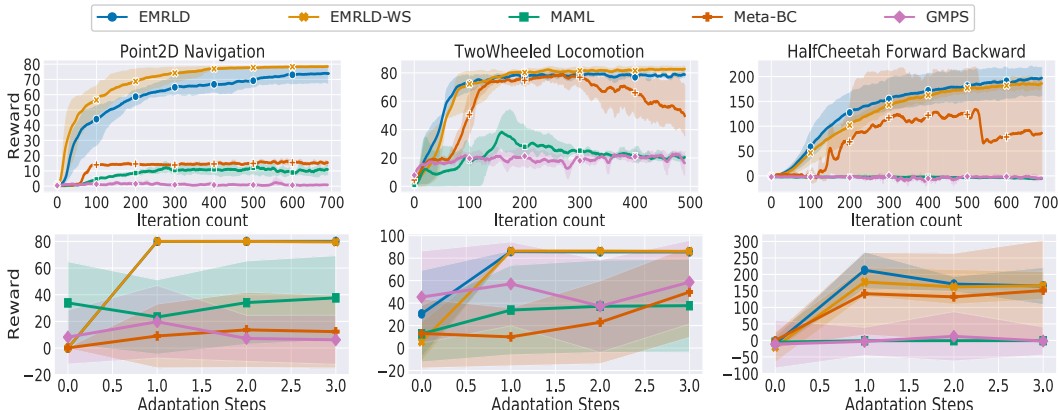

Figure 3: Training (top) and test (bottom) plots on 2D Navigation, Wheeled locomotion and Half Cheetah with **optimal demonstration data**. For training curves, a solid line corresponds to the mean over 3 seeds and the shaded region corresponds to the standard deviation over them. For the test plots, a solid line corresponds to the mean performance over all testing tasks, and the shaded region corresponds to the standard deviation over them.

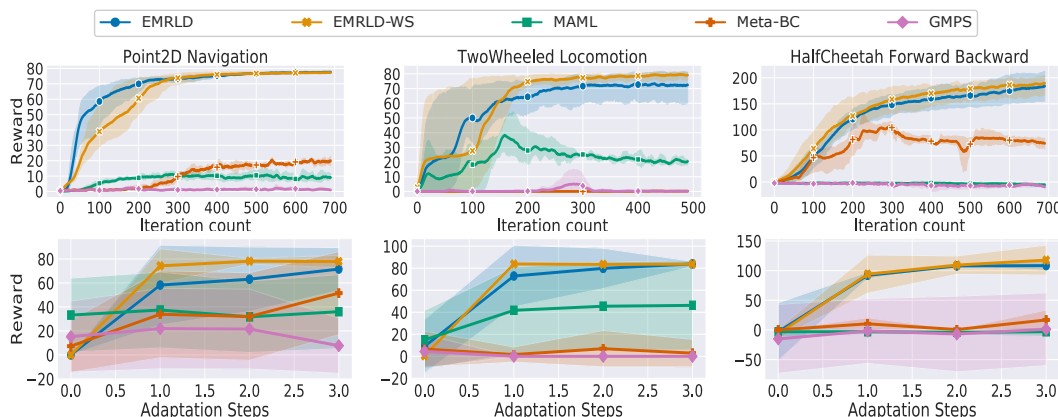

Figure 4: Training (top) and adaptation (bottom) for 2D Navigation, Wheeled locomotion and Half Cheetah with **sub-optimal demonstration data**. Notation is similar to Figure 3.

similar fashion for all tasks, but remove state-action pairs at the beginning of each trajectory. Since the first few state-action pairs contain information on how to orient the two-wheeled agent towards the goal, this truncation eliminates the possibility of direct imitation being successful. Similarly, in HalfCheetah we use a partially trained policy and truncate trajectories before they reach the reward bearing region. Figure 4 illustrates that EMRLD outperforms all the baselines and is quickly able to adapt to unseen tasks, emphasizing the benefit of its RL component. Meta-BC and GMPS fail because they are restricted by the optimality of the data, and the absence of crucial information greatly impacts their performance. MAML again fails due to the sparsity of the reward.

We conclude by presenting in Figure 5, sets of trajectories generated during testing in the TwoWheeled-Lococmotion environment when provided with optimal or sub-optimal demonstration data. The variants of EMRLD clearly outperform the others, showing their strength in the sparse reward setting.

## 4.2 Real-world Experiments on TurtleBot

We demonstrate the ability of EMRLD variants to adapt to sparse-reward tasks when they differ in environment dynamics and have sparse reward feedback. We do so via performance evaluation in the real-world using a TurtleBot shown in Figure 6 (left). We first we modify the *TwoWheeled*

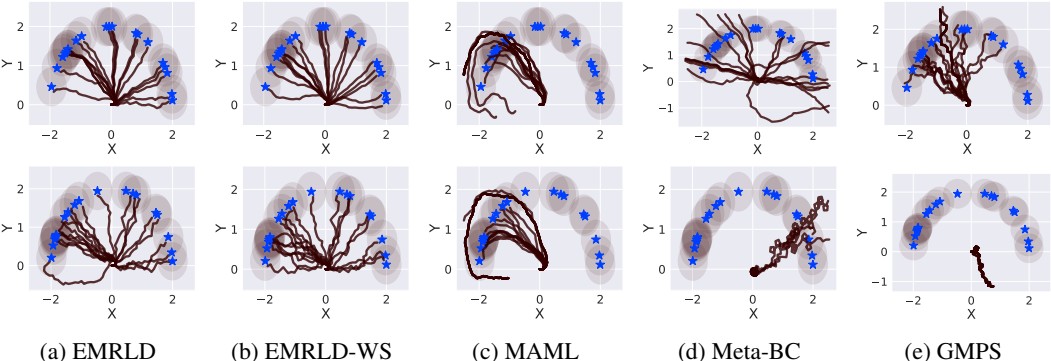

| (a) EMRLD | (b) EMRLD-WS | (c) MAML | (d) Meta-BC | (e) GMPS |

Figure 5: Trajectories after one step of adaptation using optimal demonstration data (top), and after 3 steps of adaptation using sub-optimal data (bottom) for 20 random goal points in *TwoWheeled Locomotion*. The star indicates the goal, and reward is available only in the shaded region.

*Locomotion* sparse reward environment by fixing the goal, and changing the dynamics by inducing a residual angular velocity which mimics drifts in the environment. This environmental drift is what differentiates each task. In other words, for a given task, the environment would cause the robot to drift in some specific unknown direction. We train on a set of 9 tasks with different angular velocity values (i.e., 9 different driving environments). We use one trajectory of demonstration data per task collected using an expert policy trained using TRPO. Note that all the training and data collection is done in simulation. The results are shown in Figure 6 (middle), where we see that the EMRLD variants clearly outperform the others.

For testing, we consider the environment where the Turtlebot experiences a fixed but unknown residual angular velocity representing environmental drift. Thus, we bias the angular velocity control of the TurtleBot by some amount unknown to the algorithm under test. We first execute the meta policy on the TurtleBot in the real world to collect 5 trajectories. We also provide one trajectory of simulated demonstration data. We use these samples to adapt the meta-policy, and execute the adapted task-specific policy on the TurtleBot. The results are shown in Figure 6 (right), where the origin is at $(0, 0)$ and the goal is indicated by a star. It is clear that the variants of EMRLD are the best at quickly adapting to the drift in the environment and are successful with just one step of adaptation.

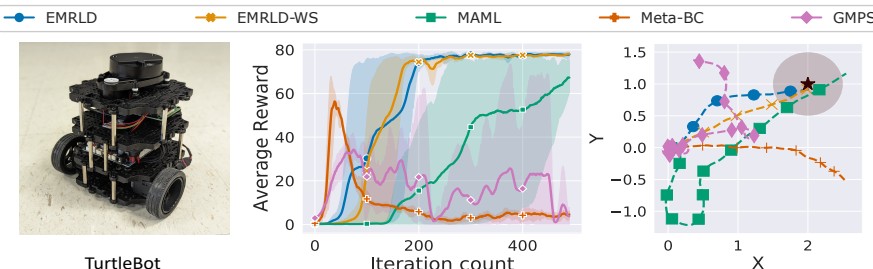

Figure 6: The TurtleBot (left) is a small, two-wheeled robot. We train the algorithms in simulation (middle) and obtain trained meta-policies. The meta-policies are adapted under an unknown drift in the real world to control the TurtleBot to attain a given goal point (right).

## 5   Conclusion

We studied the problem of meta-RL algorithm design for sparse-reward problems, in which demonstration data generated by a possibly inexpert policy is also provided. Our key observation was that simple application of an untrained meta-policy in a sparse-reward environment might not provide meaningful samples, and guidance provided by imitating the inexpert policy can greatly assuage this effect. We first showed analytically that this insight is accurate and that meta-policy improvement might be feasible as long as the inexpert demonstration policy has an advantage. We then developed two meta-RL algorithms, EMRLD and EMRLD-WS that are enhanced by using demonstration data.

We show through extensive simulations, as well as real world robot experiments that EMRLD is able to offer a considerable advantage over existing approaches in sparse reward scenarios.

## 6  Limitations and Future Work

EMRLD inherits the limitations of the gradient-based meta-RL approaches like MAML namely on-policy training, and data collection and gradient computation during test adaptation. A limitation specific to our proposed algorithms is the assumption on availability of task specific demonstration data. However, we reiterate that for a small number of train tasks, this assumption is quite practical, further, our framework allows for this data to be sub-optimal.

A possible future direction to explore is the context based meta-RL (that does't require gradient computation during testing) with demonstration data. Another future work direction is to explore usage of demonstration data in off-policy meta-RL algorithms.

## 7  Acknowledgement

This work was supported in part by the National Science Foundation (NSF) grants NSF-CAREER-EPCN-2045783 and NSF ECCS 2038963, and U.S. Army Research Office (ARO) grant W911NF-19-1-0367. Any opinions, findings, and conclusions or recommendations expressed in this material are those of the authors and do not necessarily reflect the views of the sponsoring agencies.

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

## Ethics Statement and Societal Impacts

Our work considers the theory and instantiation of meta-RL algorithms that were trained and tested on simulation and robot environments. No human subjects or human generated data were involved. Thus, we do not perceive ethical concerns with our research approach.

While reinforcement learning shows much promise for application to societally valuable systems, applying it to environments that include human interaction must proceed with caution. This is because guarantees are probabilistic, and ensuring that the risk is kept within acceptable limits is a must to ensure safe deployments.

