# Supplementary material: Enhanced Meta Reinforcement Learning using Demonstrations in Sparse Reward Environments

## A  Proof of Theorem 3.2

We will use the well known Performance Difference Lemma [16] in our analysis.

**Lemma A.1** (Performance difference lemma, [16]). *For policies any two policies $\pi_1$ and $\pi_2$,*

$$J(\pi_1) - J(\pi_2) = \frac{1}{1-\gamma}\mathbb{E}_{s\sim d^{\pi_1},a\sim\pi_1(s,\cdot)}\left[A^{\pi_2}(s,a)\right], \tag{1}$$

*where $J(\pi_j) = \mathbb{E}_{s_0\sim\rho}\left[V^{\pi_j}(s_0)\right] = \frac{1}{(1-\gamma)}\mathbb{E}_{s\sim d^{\pi_j},a\sim\pi_j(s,\cdot)}\left[R(s,a)\right],$ for $j = 1,2.$*

*Proof of Theorem 3.2.* Recall the following notations: $\pi_k$ is the meta-policy used at iteration $k$ of our algorithm, $\pi_{k,i}$ is the policy obtained after task-specific adaptation for task $i$, $d_i^{\pi_{k,i}}$ is the state-visitation frequency of policy $\pi_{k,i}$ for task $i$, and $J_i(\pi_{k,i})$ is the value of the policy for the MDP corresponding to task $i$. The value of the meta-policy $\pi_k$ is defined as $J_{\text{meta}}(\pi_k) = \mathbb{E}_{i\sim p(\mathcal{T})}[J_i(\pi_{k,i})].$

We can obtain a performance difference lemma for the meta-policies as follows.

$$
\begin{aligned}
(1-\gamma)\left(J_{\text{meta}}(\pi_{k+1}) - J_{\text{meta}}(\pi_k)\right) &= (1-\gamma)\left(\mathbb{E}_{i\sim p(\mathcal{T})}\left[J_i(\pi_{k+1,i})\right] - \mathbb{E}_{i\sim p(\mathcal{T})}\left[J_i(\pi_{k,i})\right]\right)\\
&= (1-\gamma)\mathbb{E}_{i\sim p}\left[J_i(\pi_{k+1,i}) - J_i(\pi_{k,i})\right]\\
&= \mathbb{E}_{i\sim p(\mathcal{T})}\left[\mathbb{E}_{s\sim d_i^{\pi_{k+1,i}},a\sim\pi_{k+1,i}(s,\cdot)}\left[A_i^{\pi_{k,i}}(s,a)\right]\right]\\
&= \mathbb{E}_{i\sim p(\mathcal{T}),s\sim d_i^{\pi_{k+1,i}},a\sim\pi_{1,i}(s,\cdot)}\left[A_i^{\pi_{k,i}}(s,a)\right], \tag{2}
\end{aligned}
$$

where the third equality follows from Lemma A.1. Staring from (2), we get

$$
\begin{aligned}
(1-\gamma)\left(J_{\text{meta}}(\pi_{k+1}) - J_{\text{meta}}(\pi_k)\right) &= \mathbb{E}_{i\sim p(\mathcal{T}),s\sim d_i^{\pi_{k+1,i}},a\sim\pi_{k+1,i}(s,\cdot)}\left[A_i^{\pi_{k,i}}(s,a)\right]\\
&- \mathbb{E}_{i\sim p(\mathcal{T}),s\sim d_i^{\pi_{k,i}},a\sim\pi_{k+1,i}(s,\cdot)}\left[A_i^{\pi_{k,i}}(s,a)\right] + \mathbb{E}_{i\sim p(\mathcal{T}),s\sim d_i^{\pi_{k,i}},a\sim\pi_{k+1,i}(s,\cdot)}\left[A_i^{\pi_{k,i}}(s,a)\right]\\
&= \sum_i p(i)\sum_s d_i^{\pi_{k,i}}(s)\sum_a \pi_{k+1,i}(s,a)A_i^{\pi_{k,i}}(s,a)\\
&\quad + \sum_i p(i)\sum_s \left(d_i^{\pi_{k+1,i}}(s) - d_i^{\pi_{k,i}}(s)\right)\sum_a \pi_{k+1,i}(s,a)A_i^{\pi_{k,i}}(s,a) \tag{3}
\end{aligned}
$$

We with now consider the second term of equation 3:

$$\sum_i p(i) \sum_s \left(d_i^{\pi_{k+1},i}(s) - d_i^{\pi_k,i}(s)\right) \sum_a \pi_{k+1,i}(s,a) A_i^{\pi_k,i}(s,a)$$

$$= \sum_i p(i) \sum_s d_i^{\pi_{k+1},i}(s) \sum_a \pi_i^{\text{dem}}(s,a) A_i^{\pi_2,i}(s,a)$$

$$+ \sum_i p(i) \sum_s d_i^{\pi_{k+1},i}(s) \sum_a \left(\pi_{k+1,i}(s,a) - \pi_i^{\text{dem}}(s,a)\right) A_i^{\pi_k,i}(s,a)$$

$$- \sum_i p(i) \sum_s d_i^{\pi_k,i}(s) \sum_a \left(\pi_{k+1,i}(s,a) - \pi_{k,i}(s,a)\right) A_i^{\pi_k,i}(s,a)$$

$$- \sum_i p(i) \sum_s d_i^{\pi_k,i}(s) \sum_a \pi_{k,i}(s,a) A_i^{\pi_k,i}(s,a)$$

$$\overset{(a)}{\geq} \Delta - \sum_i p(i) \sum_s d_i^{\pi_{k+1},i}(s) \sum_a \left|\pi_{k+1,i}(s,a) - \pi_i^{\text{dem}}(s,a)\right| \left|A_i^{\pi_k,i}(s,a)\right|$$

$$- \sum_i p(i) \sum_s d_i^{\pi_k,i}(s) \sum_a \left|\pi_{k+1,i}(s,a) - \pi_{k,i}(s,a)\right| \left|A_i^{\pi_k,i}(s,a)\right|$$

$$\overset{(b)}{=} \Delta - 2C_1 \mathbb{E}_{i \sim p(\mathcal{T})} \left[D_{TV}^{\pi_{k+1},i}\left(\pi_{k+1,i}, \pi_i^{\text{dem}}\right)\right] - 2C_1 \mathbb{E}_{i \sim p(\mathcal{T})} \left[D_{TV}^{\pi_k,i}\left(\pi_{k+1,i}, \pi_{k,i}\right)\right] \quad (4)$$

Here, we get $(a)$ is from Assumption 3.1 from which we have $\sum_a \pi_i^{\text{dem}}(s,a) A_i^{\pi_k,i}(s,a) \geq \Delta, \ \forall s, i,$ and noting that $\sum_a \pi_{k,i}(s,a) A^{\pi_k,i}(s,a) = 0$ by definition of advantage function. We get $(b)$ by denoting $C_1 = \max_i \max_{s,a} \left|A_i^{\pi_k,i}(s,a)\right|$. Using (4) in 3, we get

$$J_{\text{meta}}(\pi_{k+1}) - J_{\text{meta}}(\pi_k) \geq \left(\frac{1}{1-\gamma} \mathbb{E}_{i \sim p(\mathcal{T}),(s,a) \sim d_i^{\pi_k,i}} \left[\frac{\pi_{k+1,i}(s,a)}{\pi_{k,i}(s,a)} A_i^{\pi_k,i}(s,a)\right]\right.$$

$$\left. - \frac{2C_1}{1-\gamma} \mathbb{E}_{i \sim p(\mathcal{T})} \left[D_{TV}^{\pi_k,i}\left(\pi_{k+1,i}, \pi_{k,i}\right)\right]\right) + \left(\frac{\Delta}{1-\gamma} - \frac{2C_1}{1-\gamma} \mathbb{E}_{i \sim p(\mathcal{T})} \left[D_{TV}^{\pi_{k+1},i}\left(\pi_{k+1,i}, \pi_i^{\text{dem}}\right)\right]\right),$$

which completes the proof. □

# B  Environments

In this section, we describe all the simulation and real-world environments in detail.

## B.1  Simulation Environments

**Point 2D Navigation:** Point 2D Navigation [9] is a 2 dimensional goal reaching environment with $\mathcal{S} \subset \mathbb{R}^2, \mathcal{A} \subset \mathbb{R}^2$, and the following dynamics,

$$x_{t+1} = x_t + dx_t, \quad y_{t+1} = x_t + dy_t, \quad \text{such that } dx_t^2 + dy_t^2 \leq 0.1^2$$

Where $x_t$ and $y_t$ are the $x$ and $y$ location of the agent, $dx_t$ and $dy_t$ are the actions taken which correspond to the displacement in the $x$ and $y$ direction respectively, all taken at time step $t$. The goals are located on a semi circle of radius 2, and the episode terminates when the agent reaches the goal or spends more than 100 time steps in the environment. The sparse reward function for the agent is defined as follows,

$$R_t = \begin{cases} 1 - \sqrt{(x_{t+1} - x_g)^2 + (y_{t+1} - y_g)^2} & \text{if } \sqrt{(x_{t+1} - x_g)^2 + (y_{t+1} - y_g)^2} \leq 0.2 \\ 100 - t - 1 & \text{if } \sqrt{(x_{t+1} - x_g)^2 + (y_{t+1} - y_g)^2} \leq 0.02, \\ 0 & \text{otherwise,} \end{cases}$$

where $x_g$ and $y_g$ are the $x, y$ location of the goal. The agent is given a zero reward everywhere except when it is a certain distance $D_1 = 0.2$ near the goal location. Within the distance $D_1$, the agent is given two kinds of rewards. If the agent is very close to the goal, say a distance $D_2 = 0.02$, then it rewarded with a positive bonus of $1 \times$ `Number_of_times_steps_remaining_in_episode`. This

is done to create a sink near goal location to trap the agent inside it, rather than letting it wander in the $D_1$ region to keep collecting misleading positive reward. For distances between 0.02 and 0.2, the agent is given a positive reward of 1-dist(agent,goal).

**TwoWheeled Locomotion:** The TwoWheeled Locomotion environment [12] is designed based on the two wheeled differential drive model with $\mathcal{S} \subset \mathbb{R}^2$, $\mathcal{A} \subset \mathbb{R}^2$, and the following dynamics,

$$x_{t+1} = x_t + v_t \cos(\theta_t)dT, \ \ y_{t+1} = y_t + v_t \sin(\theta_t)dT, \ \ \theta_{t+1} = \theta_t + \omega_t dT,$$

with $v_t \in [0, 0.22], \omega_t \in [-2.84, 2.84]$, where $x_t, y_t$ correspond to the $x$ and $y$ coordinate of the agent, $v_t$ and $\omega_t$ are the actions corresponding to the linear and angular velocity of the agent all at time $t$, and $dT = 0.5$ is the time discretization factor. Goals are located on a semi-circle of radius 2, and the episode terminates if the agent reaches the goal, or spends more than 100 time steps in the environment, or moves out of region, which is a square box of side 2.5. The sparse reward function for the agent is defined as follows,

$$R_t = \begin{cases} 1 - \sqrt{(x_{t+1} - x_g)^2 + (y_{t+1} - y_g)^2} & \text{if } \sqrt{(x_{t+1} - x_g)^2 + (y_{t+1} - y_g)^2} \leq 0.5 \\ 100 - t - 1 & \text{if } |x_{t+1} - x_g| \leq 0.2 \quad \text{and} \quad |y_{t+1} - y_g| \leq 0.2, \\ 0 & \text{otherwise,} \end{cases}$$

where $x_g$ and $y_g$ are the $x, y$ location of the goal.

**Half Cheetah Forward-Backward:** The Half Cheetah Forward-Backward environment [9], is a modified version of the standard MuJoCo[35] HalfCheetah environment with $\mathcal{S} \subset \mathbb{R}^{20}$ and $\mathcal{A} \subset \mathbb{R}^6$, where the agent is tasked with moving forward or backward, with the episode terminating if the agent spends more than 100 time steps in the environment. The sparse reward function is as follows,

$$R_t = \begin{cases} d_g \cdot \dfrac{(x_{t+1} - x_t)}{dT} - c_t & \text{if } |x_{t+1} - x_0| > 2. \\ 0 & \text{otherwise,} \end{cases}$$

where $x_t$ corresponds to the $x$ position of the agent, $c_t$ is the control cost, all at time step $t$, $dT$ is the time discretization factor, and $d_g$ is the goal direction, which is $+1$ for the forward task and $-1$ for the backward task.

**TwoWheeled Locomotion - Changing Dynamics:** We modify the TwoWheeled locomotion environment by fixing the goal to $(2, 1)$, and adding a residual angular velocity,

$$x_{t+1} = x_t + v_t \cos(\theta_t)dT, \quad y_{t+1} = y_t + v_t \sin(\theta_t)dT, \quad \theta_{t+1} = \theta_t + \omega_g + \omega_t dT$$

with $v_t \in [0, 0.15], \omega_t \in [-1.5, 1.5]$, where $\omega_g$ is the residual angular velocity, which corresponds to different task, and mimics drift in the environment. The sparse reward function is similar to the one described in section B.1.

$$R_t = \begin{cases} 1 - \sqrt{(x_{t+1} - 2)^2 + (y_{t+1} - 1)^2} & \text{if } \sqrt{(x_{t+1} - x_g)^2 + (y_{t+1} - y_g)^2} \leq 0.5 \\ 100 - t - 1 & \text{if } |x_{t+1} - 2| \leq 0.1 \quad and \quad |y_{t+1} - 1| \leq 0.1, \\ 0 & \text{otherwise,} \end{cases}$$

### B.2 Real-World TurtleBot Platform and Experiments

We deploy the policy trained on the environment described in section B.1 on a TurtleBot 3 [2], a real world open source differential drive robot. We use ROS as a middleware to set up communication between the bot and a custom built OpenAI Gym environment. The OpenAI Gym environment acts as an interface between the policy being deployed and the bot. The custom built environment, subscribes to ROS topics (/odom for $x_t, y_t, \theta_t$), which are used to communicate the state of the bot, and publish (/cmd_vel for $v_t, \omega_t$) actions. This is done asynchronously through a callback driven mechanism. The bot transmits its state information over a wireless network to an Intel NUC, which transmits back the corresponding action according to the policy being deployed. The trajectories executed by the adapted policies are plotted in figure 1 ( note that figure 1 is the same as figure 6, re-plotted here for clarity). During policy execution on the TurtleBot, we set the residual angular velocity that mimics drift to $\omega_g = -0.65$, we note that our algorithms (EMRLD and EMRLD-WS) are able to adapt to the drift in the environment and reach the goal. We further note that MAML, takes a longer sub-optimal route to reach the reward region, but misses the goal.

We have provided a link to real-world demonstration with our code[1]. For EMRLD, we show the execution of the meta policy used to collect data, and the adapted policy. It can be clearly seen that the the meta policy collects rewards in the vicinity of the goal region, which is then used for adaptation. The adapted policy then reaches the goal. We further show the execution of the adapted policies for the baseline algorithms on the TurtleBot, and we can observe that EMRLD and EMRLD-WS outperform all the baseline algorithms and reach the goal.

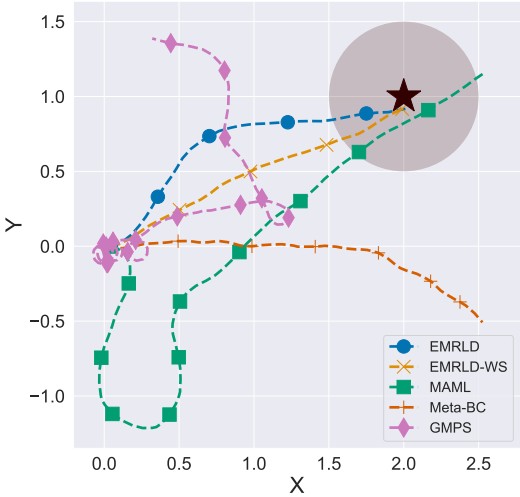

Figure 1: Trajectories in the real world for all algorithms with residual angular velocity $\omega_g = -0.65$

## C   Experimental Setup

**Computing infrastructure and run time:** The experiments are run on computers with AMD Ryzen Threadripper 3960X 24-Core Processor with max CPU speed of 3800MHz. Our implementation does not make use of GPUs. Insead, the implementation is CPU thread intensive. On an average, EMRLD and EMRLD-WS take ~3h to run on smaller environments, and take ~5h on HalfCheetah. We train goal conditioned expert policies using TRPO. Expert policy training takes ~0.5h to run. Our code is based on *learn2learn*[2] [3], a software library built using PyTorch [24] for Meta-RL research.

**Neural Network and Hyperparameters:** In our work, the meta policy $\pi^{\text{dem}}$ and the adapted policies $\pi_{k,i}$ are stochastic Gaussian policies parameterized by neural networks. The input for each policy network is the state vector $s$ and the output is a Gaussian mean vector $\mu$. The standard deviation $\sigma$ is kept fixed, and is not learnable. During training, an action is sampled from $\mathcal{N}(\mu, \sigma)$.

For value baseline (used for advantage computation) of meta-learning algorithms, we use a linear baseline function of the form $B(s) = \zeta_{s,t}^\top G(s)$, where $\zeta_{s,t} =$ concat $\left(s, s \odot s, 0.01t, (0.01t)^2, (0.01t)^3, 1\right)$, and $G(s)$ is discounted sum of rewards starting from state $s$ till the end of an episode. This was first proposed in [7] and is used in MAML [9]. This is preferred as a learnable baseline can add additional gradient computation and backpropagation overheads in meta-learning.

We use TRPO on goal conditioned policies to obtain optimal and sub-optimal experts for all the tasks in an environment at once. For each environment, the task context variable, *i.e.*, a vector that contains differentiating information on a task, is appended to the input state vector of the policy network. The rest of the policy mechanism is same as described above for meta-policies. A learnable value network is used to cut variance in advantage estimation. Once the expert policy is trained to the desired amount, just one trajectory per task is sampled to construct demonstration data.

All the models used in this work are multi-layer perceptrons (MLPs). The policy models for all the meta-learning algorithms have two layers of 100 neurons each with Rectified Linear Unit (ReLU) non-linearities. The data generating policy and value models use two layers of 128 neurons each.

Table 1 lists the set of hyperparameters used for EMRLD, EMRLD-WS and the baseline algorithms. In addition to the ones listed in Table 1, meta batch size is dependant on the training environment: it is 12 for Point2D Navigation, 24 for TwoWheeled Locomotion and 10 for HalfCheetah Forward-Backward. In Table 1, Meta LR specified as 'TRPO' means that the learning rate is determined by step-size rule coming from TRPO. The meta optimization steps in Meta-BC and GMPS use ADAM [18] optimizer with a learning rate of 0.01. We use 20 CPU cores to parallelize policy rollouts for adaptation. The hyperparameters $w_{\text{rl}}$ and $w_{\text{bc}}$ are kept fixed across environments for EMRLD and EMRLD-WS. The parameter $w_{\text{bc}}$ is kept at 1 for both optimal and sub-optimal data, and across

---

[1]https://github.com/DesikRengarajan/EMRLD

[2]https://github.com/learnables/learn2learn

environments. The parameter $w_{rl}$ takes a lower value of $0.2$ across environments for optimal data as in practise optimal data is expected to be highly informative. Hence, we desire the gradient component arising from optimal data to hold more value while adaptation. For sub-optimal data, the agent is required to explore to obtain performance beyond data, and hence, $w_{rl}$ is kept at 1. We further show in section D that our algorithm is robust to choice of $w_{bc}$ and $w_{rl}$.

| HYPERPARAMETER | EMRLD | EMRLD-WS | MAML | META-BC | GMPS |
|---|---|---|---|---|---|
| ADAPTATION LR | 0.01 | 0.01 | 0.01 | 0.01 | 0.01 |
| META LR | TRPO | TRPO | TRPO | 0.01(ADAM) | 0.01(ADAM) |
| ADAPT STEPS | 1 | 1 | 1 | 1 | 1 |
| ADAPT BATCH SIZE | 20 | 20 | 20 | 20 | 20 |
| GAE $\tau$ | 1 | 1 | 1 | 1 | 1 |
| $\gamma$ | 0.95 | 0.95 | 0.95 | 0.95 | 0.95 |
| CPU THREAD NO. | 20 | 20 | 20 | 20 | 20 |
| $w_{rl}$ | 0.2/1 | 0.2/1 | N/A | N/A | N/A |
| $w_{bc}$ | 1 | 1 | N/A | N/A | N/A |

Table 1: Hyperparameter values for EMRLD, EMRLD-WS, MAML, Meta-BC and GMPS. The hyperparameters are kept fixed across algorithms, across environments, and across demonstration data type.

## D  Sensitivity Analysis

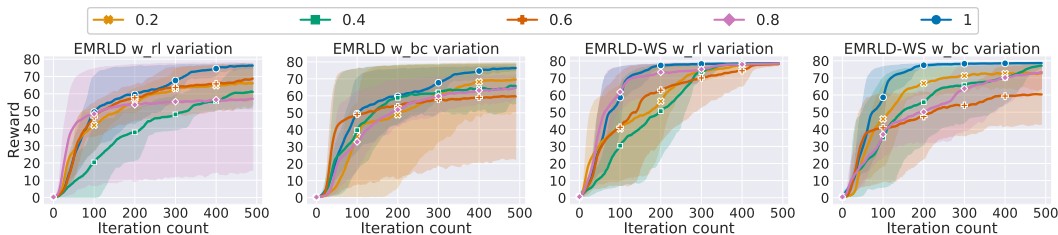

Figure 2: Sensitivity analysis for EMRLD and EMRLD-WS when the demonstration data is optimal. (a) Keeping $w_{bc} = 1$, $w_{rl}$ is varied from $0.2$ to $1$ in steps of $0.2$ for EMRLD. (b) Keeping $w_{rl} = 1$, $w_{bc}$ is varied from $0.2$ to $1$ in steps of $0.2$ for EMRLD. (c) Keeping $w_{bc} = 1$, $w_{rl}$ is varied from $0.2$ to $1$ in steps of $0.2$ for EMRLD-WS. (d) Keeping $w_{rl} = 1$, $w_{bc}$ is varied from $0.2$ to $1$ in steps of $0.2$ for EMRLD-WS.

We perform sensitivity analysis for parameters $w_{rl}$ and $w_{bc}$ on our algorithms EMRLD and EMRLD-WS for optimal data on Point2D Navigation. The results for the same are included in Fig. 2. All the plots are averaged over three random seed runs. To assess the sensitivity of our algorithms to $w_{rl}$, we fix $w_{bc} = 1$ and vary $w_{rl}$ to take values from $0.2, 0.4, 0.6, 0.8$ and $1$. Similarly, to assess how sensitive our algorithm's performance to $w_{bc}$ is, we fix $w_{rl} = 1$ and vary $w_{bc}$ to take values from $0.2, 0.4, 0.6, 0.8$ and $1$. All the hyperparameters are kept fixed to the values listed in Table 1. We observe that our algorithms are fairly robust to variations in $w_{rl}$ and $w_{bc}$ for three random seeds. Since demonstration data is leveraged to extract useful information regarding the environment and the reward structure, our algorithms are slightly more sensitive to $w_{bc}$ variation than $w_{rl}$ variation.

## E  Ablation experiments

We perform ablation experiments for EMRLD by setting $w_{bc} = 0$ and $w_{rl} = 0$ on the Point2D Navigation environment with the optimal and the sub-optimal demonstration data. We observe from figure 3, that setting $w_{bc} = 0$ hampers the performance to a greater extant as the agent is unable to extract useful information from the environment due to the sparse reward structure. We also observe that setting $w_{rl} = 0$ hampers the performance, as the agent is unable to exploit the RL structure of the problem to achieve high rewards.

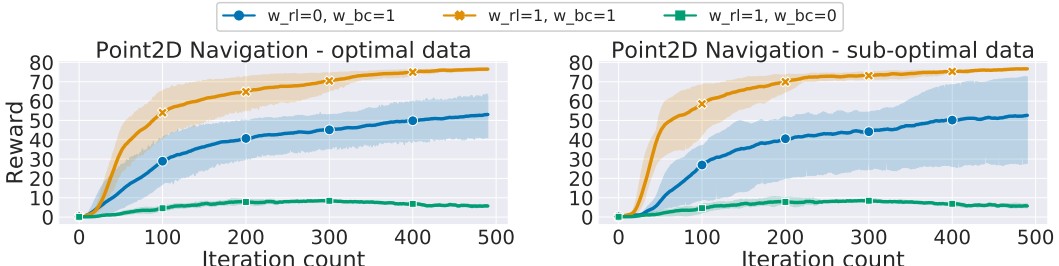

Figure 3: Ablation experiments on EMRLD for Point2D Navigation environment by changing $w_{bc}$ and $w_{rl}$ in the adaptation step with the optimal and the sub-optimal demonstration data.

# F  Related Work

**Meta-Learning:**  Reinforcement learning (RL) has become popular as a tool to perform *learning from interaction* in complex problem domains like autonomous navigation of stratospheric balloons [5] and autonomously solving a game of Go [32]. In large scale complex environments, one requires a large amount of data to learn any meaningful RL policy [6]. This is in stark contrast to how we as humans behave and learn - by translating our *prior knowledge* of past exposure to same/similar tasks into behavioural policies for a new task at hand. The initial work [30] took to addressing the above mentioned gap and proposed the paradigm of meta-learning. The idea has been extended to obtain gradient based algorithms in supervised learning, unsupervised learning, control, and reinforcement learning [31, 15, 34, 37, 8]. More recently, model-agnostic meta-learning (MAML) [9] introduced a gradient based two-step approach to meta-learning: an inner adaptation step to learn specific task policies, and an outer meta-optimization loop that implicitly makes use of the inner policies. MAML can be used both in the supervised learning and RL contexts. Reptile [22] introduced efficient first order meta-learning algorithms. PEARL [26] takes a different approach to meta-RL, wherein task specific contexts are learned during training, and interpreted from trajectories during testing to solve the task. In its native form, the RL variant of MAML can suffer from issues of inefficient gradient estimation, exploration, and dependence on a rich reward function. Among others, algorithms like ProMP [28] and DiCE [11] address the issue of inefficient gradient estimation. Similarly, E-MAML [1, 33] and MAESN [12] deal with the issue of exploration in meta-RL. Inadequate reward information or sparse rewards is a particularly challenging problem setting for RL , and hence, for meta-RL. Very recently, HTR [23] proposed to relabel the experience replay data of any off-policy algorithm to overcome exploration difficulties in sparse reward goal reaching environments. Different from this approach, we leverage the popular *learning from demonstration* idea to aid learning of meta-policies on tasks including and beyond goal reaching ones.

**RL with demonstration:** 'Learning from demonstrations' (LfD) [29] first proposed the use of demonstrations in RL to speed up learning. Since then, leveraging demonstrations has become an attractive approach to aid learning [13, 36, 21]. Earlier work has incorporated data from both expert and inexpert policies to assist with policy learning in sparse reward environments [21, 14, 36, 17, 27]. In particular, DQfD [14] utilizes demonstration data by adding it to the replay buffer for Q-learning. DDPGfD[36] extend use of demonstration data to continuous action spaces, and is built upon DDPG [19]. DAPG [25] proposes an online fine-tuning algorithm by combining policy gradient and behavior cloning. POfD [17] propose an approach to use demonstration data through an appropriate loss function into the RL policy optimization step to implicitly reshape sparse reward function. LOGO [27] proposes a two-step guidance approach where demonstration data is used to guide the RL policy in the initial phase of learning.

**Meta-RL with demonstration:** Use of demonstration data in meta-RL is new, and the works in this area are rather few. Meta Imitation Learning [10] extends MAML [9] to imitation learning from expert video demonstrations. WTL [39] uses demonstrations to generate an exploration algorithm, and uses the exploration data along with demonstration data to solve the task. ODA [38] use demonstration data to perform offline meta-RL for industrial insertion, and [4] propose generalized 'upside down RL' algorithms that use demonstration data to perform offline-meta-RL. GMPS [20] extends MAML [9] to leverage expert demonstration data by performing meta-policy optimization via supervised learning.

Closest to our approach are GMPS [20] and Meta Imitation Learning [10], and we will focus on comparisons with versions of these algorithms, along with the original MAML [9].