# OpenReview forum: "Enhanced Meta Reinforcement Learning via Demonstrations in Sparse Reward Environments"
_NeurIPS.cc/2022/Conference — NeurIPS 2022 Accept_

### Official Review · Reviewer_9vCH · 2022-07-06

**Rating:** 7
**Confidence:** 4
**Soundness:** 3 good
**Presentation:** 3 good
**Contribution:** 3 good

**Summary:**

The paper introduces a new method (EMRLD) for meta reinforcement learning with a focus on sparse reward environments and learning from sub-optimal demonstrations. The authors first introduce meta reinforcement learning and the limitations of approaches like MAML in sparse reward settings, as well as relevant work in reinforcement learning and meta reinforcement learning with demonstrations. Subsequently, the authors describe the problem of learning from suboptimal demonstrations in meta-RL in greater detail and introduce their loss formulation and general algorithm (EMRLD) whose structure is inspired by MAML and makes use of combined loss from RL and supervised learning (behavioral cloning) to learn from sub-optimal demonstrations. The authors then present a theorem on how sub-optimal demonstrations can aid EMRLD in achieving superior performance to gradient-based algorithms that do not use demonstration data and then showcase how their method compares to several baselines in different experiments.

The experiments generally show that EMRLD perform as well or better compared to the baseline methods in settings with optimal demonstrations and significantly better in settings where demonstration data is suboptimal in the simulated environments. The authors then showcase results for a TurtleBot navigation setting where the robot is trying to reach a given goal.

**Questions:**

**General Questions:**
- Could you clarify the general architectural setup that you are using? Based on the formulation it appears that the meta-policy and task policies share parameters and it would be good to have more clarity and how the overall neural network architectures express that.
- How do you determine how many task specific policies you use? Is this hyperparameter or is there a general method?
- In the EMRL-WS setting, how do determine when to stop the BC warm start?
- Can you provide more clarity on how the variance in the bottom of Figure 3 and Figure 4 (adaptation setting) affects your reasoning about how well your method performs against the baselines? The sub-optimal demonstration setting shows a clearer difference between the methods compared to the expert demonstration setting, but it would still be good to get more clarity on this.
- Have you thought about doing transferring and /or meta-learning across different environment settings? This might be worth thinking about for future work.

**Limitations:**

I think the paper could be improved by a more thorough discussion of the limitations of EMRLD as well as future work that it can inspire. Given space limitations, it would be OK to include this in the appendix even though I would recommend it to be in the main paper.

**Strengths And Weaknesses:**

**Originality:**
- Strengths: The paper proposes a new method for meta reinforcement learning in sparse reward settings that can leverage sub-optimal demonstrations.
- Weaknesses: The authors could have provided more detail on the limitations of their method (discussed below as well).

**Quality:**
- Strengths: The methods and contributions are generally well supported in the experiments and discussed in detail in the paper.
- Weaknesses: The authors do not address the variance in their results in Figure 3 and Figure 4 bottom plots where adaptation is shown. The variance in some of the experiments makes it difficult to determine which method performs better compared to others.

**Clarity:**
- Strengths: The paper is generally well written and well organized with relevant equations, diagrams and descriptions. The theorem provided in Section 3 appears well formulated.
- Weaknesses: To enhance readability Figure 3 and Figure 4 could use bolded "subtitles" such as "Expert Demonstrations" and "Sub-Optimal Demonstrations" to make the difference between them clear.

**Significance:**
- Strengths: The paper proposes a new method in a relevant subject area of meta reinforcement learning.
- Weaknesses: The authors could have provided more detail on the limitations of their method and put it into the broader context of meta reinforcement learning methods (discussed below as well).

---

> ### Author Response · Authors · 2022-08-02
> **Response to Reviewer 9vCH**
>
> We thank the reviewer for their thoughtful comments  on our work. We are encouraged to know that the reviewer finds our paper proposes a new method in the area of meta reinforcement learning, the contributions are well supported by experiments, and the paper is well written and organized. We have added a section on limitations and future work in Appendix H, and changed plot descriptions for clarity. Below we address the questions posed by the reviewer.
>
> **Q1.** *``The authors do not address the variance in their results in Figure 3 and Figure 4 bottom plots where adaptation is shown. The variance in some of the experiments makes it difficult to determine which method performs better compared to others.''*
>
> **Response:**  We would like to point out that the baseline algorithms (GMPS, Meta-BC, MAML) have a large standard deviation during testing as they succeed only on a small number of tasks, while both our algorithms (EMRLD, EMRLD-WS) have low standard deviation during testing as they succeed in all the tasks.  This is why the figures appear to show larger variance when the plots are all superimposed.
>
> For example, in Point2D Navigation environment, during testing, we sample 20 random target locations that are not present during training and evaluate the performance of all algorithms on these set of tasks and plot the mean and standard deviation of their performance. The baseline algorithms have a high standard deviation since the agent reaches only some of the 20 random targets, this can be seen in the trajectory plots in Figure 5. Different from this,  both EMRLD and EMRLD-WS reach all of the targets resulting in a lower standard deviation.
>
> **Q2.** *``The authors could have provided more detail on the limitations of their method and put it into the broader context of meta reinforcement learning methods.''*
>
> **Response:**  We thank the reviewer for their suggestion. We have now added a section discussing limitations and future work in greater detail in the Appendix H.  We have provided more context on meta-learning dependence on on-policy training and our need for demonstration data.
>
> **Q3.** *``Could you clarify the general architectural setup that you are using?''*
>
> **Response:**  We follow the usual architecture of gradient-based meta-learning algorithms such as MAML, where the meta policy and the task policies have the same neural network architecture. The task policy neural network parameters are updated starting from the initial value of the meta-policy neural network parameter. Details on the structure of the neural network are provided in Appendix C.
>
> **Q4.** *``In the EMRL-WS setting, how do determine when to stop the BC warm start?''*
>
> **Response:** We incorporate the warm starting as a part of task specific adaptation step, and  perform just one step of BC warm start.
>
> **Q5.** *``How do you determine how many task specific policies you use? Is this hyperparameter or is there a general method?''*
>
> **Response:** During training, we train on a small number of tasks, namely 12, 24, and 2 for the point2d, TwoWheeled, and HalfCheetah environments. We have specified the details in Section 4.1.  We can adapt the meta policy to any task during testing to obtain the task specific policy.
>
> **Q6.** *``Have you thought about doing transferring and /or meta-learning across different environment settings? This might be worth thinking about for future work.''*
>
> **Response:** Please note that we have already evaluated EMRLD and EMRLD-WS in the setting where tasks are different in terms of the environment dynamics, and not only in terms of the reward functions. In particular, in  Section 4.2, we have described the evaluation of our algorithms on a real-world mobile robot where each task corresponds to a different (unknown) residual angular velocity representing environmental drift. We appreciate the reviewer's comment on this possible future direction. We are indeed planning to extend our approach to more challenging environments such as off-road navigation for mobile robots where the changes in the terrain can change the resultant environment dynamics. A meta-learning based controller is the ideal approach for rapidly adapting to such changing environment scenarios.

---

> > ### Comment · Reviewer_9vCH · 2022-08-06
> > **Reply to Author Response**
> >
> > I thank the authors for their thorough reply. Most of my questions and comments were addressed to my satisfaction. I would have preferred the discussion of limitations to be in the main paper, but understand space limitations given the current content. I have adjusted my score accordingly.

---

### Official Review · Reviewer_nwsU · 2022-07-10

**Rating:** 7
**Confidence:** 4
**Soundness:** 4 excellent
**Presentation:** 4 excellent
**Contribution:** 3 good

**Summary:**

This paper presents a meta-learning algorithm Enhanced Meta-RL using Demonstrations (EMRLD) that leverages potentially suboptimal demonstrations to improve meta reinforcement learning in sparse reward environments. The proposed method assumes access to a set of optimal or suboptimal expert demonstrations, and then uses a combined BC loss on the demos with RL policy improvement loss on online rollouts for the inner loop. Then, the outer loop uses a single RL gradient step on the online rollouts. The main motivation of the method is that compared to Guided Meta-Policy Search (GMPS), EMRLD is able to (1) not be limited by the quality of the initial demonstrations since EMRLD uses an RL outer-loop instead of BC on the original demos and (2) can take advantage of online-explored successes by the meta-policy. An additional version of the method is presented where it is warm-started with BC beforehand, which is called EMRLD-WS. They demonstrate the efficacy of both versions of EMRLD on simulated and real control domains, where EMRLD outperforms all baselines especially when demonstrations are suboptimal.

**Questions:**

- One major benefit of EMRLD is that online rollouts D_i^{val} enable the meta-policy to not be constrained by the quality of the initial demonstrations. What happens if you give GMPS this similar benefit, by using online-explored successes to the D_i^{val} used for BC in GMPS?
- One advantage of EMRLD over MAML is the usage of demonstrations. Do you allow MAML to warm-start on demonstrations or incorporate them during training naively at all?
- IL+RL is a well-studied field that has found the naive weighting of BC and RL loss to be very hyperparameter sensitive. How were w_{rl} and w_{bc} chosen? Do you have an ablation, such as when you set w_{rl} to 0.0 or when you set w_{bc} to 0.0?


**Limitations:**

The authors are proactive in mentioning the restriction to on-policy RL settings that involve increased computation cost and less data reusability.

**Strengths And Weaknesses:**

Strengths:
- The method is able to incorporate suboptimal demonstrations
- The method is not much more expensive than related methods Meta-BC and GMPS

Weaknesses:
- Fixed w_{bc} could still limit performance to be constrained around suboptimal demonstrations
- Uses more data in the outer loop compared to GMPS, so a fair baseline would be to use self-imitation in the outer loop (see question below)

---

> ### Author Response · Authors · 2022-08-02
> **Response to Reviewer nwsU**
>
> We thank the reviewer for their thoughtful comments and suggestions. Below, we provide a detailed response to the questions posed in the review.
>
> **Q1.** *``One major benefit of EMRLD is that online rollouts $D_i^{val}$ enable the meta-policy to not be constrained by the quality of the initial demonstrations. What happens if you give GMPS this similar benefit, by using online-explored successes to the $D_i^{val}$ used for BC in GMPS?.''*
>
> **Response:** We note that the inner task-specific adaptation step in GMPS uses only an RL loss. A meta-policy run in a sparse reward environment will not collect valuable rewards initially, rendering the gradients of the RL loss and policy adaptation meaningless. Hence, the validation data collected using this adapted policy may be sub-optimal. Thus, when this validation data is combined with the demonstration data to perform self-imitation for the meta update step of GMPS, it will result in a poorly performing meta policy.
>
> We simulate GMPS with self-imitation in the Point2D Navigation environment with the optimal and the sub-optimal data. We allow the adapted policy to collect the validation data, and combine the validation data with demonstration data to perform the meta update step of GMPS. We observe in Figure 9 (Appendix E) that performing self imitation does not improve the performance of GMPS.
>
> **Q2.** *``One advantage of EMRLD over MAML is the usage of demonstrations. Do you allow MAML to warm-start on demonstrations or incorporate them during training naively at all?''*
>
> **Response:**  While our experiments utilize the vanilla MAML algorithm for purposes of comparison, we would like to point out that a naïve warm start by using behavior cloning on available demonstration data to obtain a baseline policy and then continuing with MAML from this baseline policy will not result in good performance, since the cloning objective of warm start and the learn-to-lean objective of MAML are at odds with each other.  We further note that the ``warm start'' of EMRLD-WS is more nuanced, and is actually a behavior cloning step applied to the meta policy to initialize the task specific adaptation.  This enables the resultant online rollout of the policy to contain richer data with more reward information.
>
> **Q3.** *``IL+RL is a well-studied field that has found the naive weighting of BC and RL loss to be very hyperparameter sensitive. How were $w_{rl}$ and $w_{bc}$ chosen?''*
>
> **Response:** We had actually included such a sensitivity analysis in the appendix already, please see Appendix D. We note that our algorithm is  fairly robust to variations in $w_{rl}$ and $w_{bc}$.  Since demonstration data is leveraged to extract useful information regarding the environment and the reward structure, our algorithms are slightly more sensitive to $w_{bc}$ variation than $w_{rl}$ variation.
>
>
> **Q4.** *``Do you have an ablation, such as when you set $w_{rl}$ to 0.0 or when you set $w_{bc}$ to 0.0?''*
>
> **Response:** We have now performed such an ablation study. We have set $w_{rl}/w_{bc}$ to $0.0$  with optimal and sub-optimal demonstration data and included the result in the updated manuscript, please see Appendix F.  In line with our sensitivity analysis, we notice that the setting of $w_{bc}=0.0$ hampers the performance to a greater extant as the agent is unable to extract useful information from the environment due to the sparse reward structure.
>
> **Q5.** *``Fixed $w_{bc}$ could still limit performance to be constrained around sub-optimal demonstrations''*
>
> **Response:** Since we are performing only one step of adaptation during training in this work, this should not be a problem. However, as the reviewer points out, when the algorithm  performs multiple adaptation steps, decaying $w_{bc}$ may improve the performance as it will reduce the dependence on the sub-optimal demonstration data.

---

> > ### Comment · Reviewer_nwsU · 2022-08-07
> > **Thanks for the updates**
> >
> > Thank you for the clarifications and additional experiments! I have kept my positive rating.

---

### Official Review · Reviewer_y36R · 2022-07-14

**Rating:** 5
**Confidence:** 3
**Soundness:** 3 good
**Presentation:** 3 good
**Contribution:** 2 fair

**Summary:**

The paper proposes a new meta-reinforcement learning algorithm for adaptation to different tasks or environments. The algorithm targets the tasks where the reward function is extremely sparse. In these tasks the standard meta-RL algorithms fail since the reward does not give a signal for adaptation. The authors use demonstrations that can also be suboptimal. The paper shows an elegant way of integrating demonstrations using gradients, instead of doing a supervised RL. This way, the algorithm still performs well (better than demonstrations) in case the demonstrations are not optimal. The authors showcase their algorithm in various environments where rewards and environments are changed. They also apply to a 2 wheeled real robot that has differen angular velocities as environments.

**Questions:**

Please see above. Could the authors explain or give examples to the set of problems the algorithm tackles. Maybe I'm missing a point in the main motivation of the paper.

**Limitations:**

The authors cite some limitations of the algorithm, but do not discuss the concern I stated above. I think the class of problems that the algorithhm tackles is another limitation as well.

**Strengths And Weaknesses:**

The strengths of the paper is clear for the targeted class of problems where the reward is extremely sparse. The algorithm performs well for these tasks compared to other algorithms as expected. On the other hand, the other algorithms has a clear disadvantage of not having demonstrations or any guidance for these tasks with extremely sparse rewards, so the results are ver much expected.

The authors did a good job on testing the algorithm on various type of problems. I especially like that they show both the problems where the reward is changing and the problems where the environment is changing. Having a real robot experiment is also another plus.

The main weakness in my opinion is the set of problems that the authors target. They are looking at problems with extremely sparse reward, but with suboptimal demonstrations. These problems very much exist in real world for reinforcement learning. On the other hand, in the context of meta-RL and adaptation, I cannot think of a clear example where this would be needed, especially for the tasks with different rewards. We are looking into problems where the agent is put to an environment to adapt to an unknown task with a reward function that will mostly return zero unless the agent reaches to a goal that will be different for each task without explicit indicator to the agent. If the reward function for adapting is different, one could potentially train with task information as an input to the policy.  Also, if there are demonstrations, there are ways to break the sparsiticty of the problem.

In my opinion, adapting to different environments is slightly more likely to appear as a potential real world problem to solve. Even then, it might be feasible to train a robust policy (depending on the problem, of course). Or, in these problems, the authors can potentially look at the state transitions. Indeed I would suggest authors to look at the publication titled "No-Reward meta learning" NoRML, bu Yang et al.

Overall I think the authors did a good job at finding an algorithm to address the problem, present their hypothesis, test their hypothesis and presenting the results. I am only concerned with the class of problems that the algorithm tackles.

---

> ### Author Response · Authors · 2022-08-02
> **Response to Reviewer y36R**
>
> We thank the reviewer for their insightful comments and questions. We are encouraged to know that reviewer finds our methodology  ``an elegant way of integrating demonstrations using gradients, instead of doing a supervised RL", and that they appreciate our experimental section.  Below we provide a detailed response to the comments and questions posed by the  reviewer.
>
>
> **Q1.** *``The main weakness in my opinion is the set of problems that the authors target. They are looking at problems with extremely sparse reward, but with suboptimal demonstrations. These problems very much exist in real world for reinforcement learning. On the other hand, in the context of meta-RL and adaptation, I cannot think of a clear example where this would be needed, especially for the tasks with different rewards?''*
>
>
> **Response:** We believe that if a problem is important in the real-world RL, its meta-learning version is also likely to be equally important. For instance, consider the problem of using RL to obtain a policy for a robot to perform a task, where we only have a sparse reward function that only indicates whether the task is completed partially or fully.  There is much work in the RL literature that addresses the challenges of sparse rewards here [21, 14, 36, 17, 27]. Now, consider the same robot tasked with walking, running, or hopping at different target speeds, with a reward associated with attaining the target speed.  Although the dynamics remain exactly the same, changing only the reward produces a variety of tasks that naturally motivate meta-RL for such a sparse reward environment.  This is the reason why tasks that only differ in rewards are usually considered as benchmark settings in the meta-RL literature, starting from the original MAML paper [9] and going beyond [12, 23, 26]. Hence, the first set of tasks that we study are of this flavor.
>
> Additionally, we go further and consider the setting where tasks have different dynamics in the Section 4.2 of our paper, where we demonstrate the performance of our algorithm in differing real world environments using a turtle-bot. The video of the real-world demonstration is attached in the supplementary material.
>
> **Q2.** *``If the reward function for adapting is different, one could potentially train with task information as an input to the policy''*
>
> **Response:** In many settings, the task information may not be available to the learning agent to be given as an input to the policy. This is the reason why existing works on meta-RL [12, 23, 26, 20, 40], including the original MAML paper [9], proposed algorithms that do not assume that such information be provided. In our paper, we follow the same convention. We first consider the problem setting where a sparse reward is provided in a small region around an unknown goal. The agent needs to actually arrive in this region to interpret the task. We then consider the other setting where tasks differ by the virtue of dynamics (Section 4.2 and Appendix B2), where dynamics are varied by including unknown residual angular velocities representing environmental drift. Here, the agent needs to interact with the environment to interpret the task.
>
> **Q3.**  *``In my opinion, adapting to different environments is slightly more likely to appear as a potential real world problem to solve. Even then, it might be feasible to train a robust policy (depending on the problem, of course).''*
>
> **Response:** We note that a robust policy trained on multiple tasks will perform worse than a policy adapted to a particular task. This is because a robust policy is trained for worst-case or average case scenario, while a meta-policy adapted to a particular task is customized to that task.   Hence, meta-RL seems particularly valuable in the differing environment scenario.
>
> **Q4.** *``Overall I think the authors did a good job at finding an algorithm to address the problem, present their hypothesis, test their hypothesis and presenting the results. I am only concerned with the class of problems that the algorithm tackles.''*
>
> **Response:** Thank you for your appreciation of our work. We hope that our response above gives a convincing case of the importance of the class of problems our algorithm can tackle.

---

### Author Response · Authors · 2022-08-02
**General Response**

We thank all the reviewers for their thoughtful comments and feedback. We are encouraged to know that the reviewers found our work new (reviewer 9vCH), elegant (reviewer y36R), well supported with experiments (reviewer 9vCH) in simulation and the real world (reviewer y36R), and well written with relevant equations, diagrams and descriptions. The main elements of our response are as follows:

**1. Additional experiments with a variant of the baseline:** Reviewer nwsU suggested trying a version of GMPS  that uses self imitation. We have now performed this additional experiment and have  added the results in the Appendix E. We are happy to report  that our algorithm outperforms this variant as well.


**2. Ablation study:** Reviewer nwsU suggested trying an ablation study by setting $w_{bc} = 0$ and $w_{rl} = 0$.  We have now performed this additional experiment and have  added the results in the Appendix F.

**3. More discussion on limitations:** Reviewer y36R and 9vCH suggested additional discussion on the limitations of our work. We have now added a section on the limitations in the Appendix H.


Finally, we would like to point the reviewers to the **videos** in the supplementary material that demonstrate the performance of our algorithm in adapting to an environment with different dynamics in the real-world. We have provided detailed response to each reviewer's comments. We have also modified our paper to incorporate the comments and suggestions by the reviewers. The modified parts of the paper are marked in **blue**.

---

### Meta-Review · Area_Chair_GKxn · 2022-08-26

**Recommendation:** Accept
**Confidence:** Certain

**Metareview:**

The authors propose EMRLD algorithms, which use potentially suboptimal demonstrations to perform meta-RL in environments where rewards are sparse. The algorithm is illustrated well in Point2D navigation toy examples to illustrate how it solves a multi-task goal reaching environment on both suboptimal and optimal data. Empirical results on twowheeled and halfcheetah forward-backward are compelling, and I appreciated the real-world experiments on Turtlebot. All reviewers have voted to accept.

**Award:**

No

---

### Decision · Program_Chairs · 2022-09-14

Accept